# Unravelling the Complexity of Sarcopenia Through a Systems Biology Approach

**DOI:** 10.3390/ijms26178527

**Published:** 2025-09-02

**Authors:** Atakan Burak Ceyhan, Ozlem Altay, Cheng Zhang, Sehime Gulsun Temel, Hasan Turkez, Adil Mardinoglu

**Affiliations:** 1Centre for Host-Microbiome Interactions, Faculty of Dentistry, Oral & Craniofacial Sciences, King’s College London, London SE1 9RT, UK; atakan.ceyhan@kcl.ac.uk; 2Science for Life Laboratory, KTH—Royal Institute of Technology, SE-17165 Stockholm, Sweden; hoaltay@gmail.com (O.A.); cheng.zhang@scilifelab.se (C.Z.); 3Department of Medical Genetics, Faculty of Medicine, Bursa Uludag University, Bursa 16059, Turkey; sehime@uludag.edu.tr; 4Department of Translational Medicine, Institute of Health Science, Bursa Uludag University, Bursa 16059, Turkey; 5Department of Histology and Embryology, Faculty of Medicine, Bursa Uludag University, Bursa 16059, Turkey; 6Department of Medical Biology, Faculty of Medicine, Atatürk University, Erzurum 25240, Turkey; hturkez@atauni.edu.tr

**Keywords:** sarcopenia, systems biology

## Abstract

Sarcopenia, a significant loss of muscle mass and strength, is an important healthcare problem in the geriatric population. While age-related muscle decline represents the most common form, sarcopenia may also develop as a secondary condition associated with chronic diseases, including cancer, diabetes, chronic obstructive pulmonary disease, and autoimmune disorders. It increases frailty, disability, and fall risk among the elderly while also raising hospitalization rates and associated healthcare costs. Although no pharmaceutical agents have been specifically approved for the treatment of sarcopenia to date, elucidating its underlying molecular mechanisms of sarcopenia through systems biology approaches is essential for the development of novel therapeutic strategies and preventive interventions. This review examines the current definitions of sarcopenia, recent advancements in its management, and the emerging role of systems biology in uncovering potential biomarkers and therapeutic targets. We discuss how these approaches may contribute to the development of novel interventions aimed at enhancing muscle health and improving the quality of life in older adults and provide a summary of the current progress achieved through systems biology methodologies in sarcopenia research.

## 1. Introduction

Sarcopenia is a condition characterized by the progressive loss of muscle mass and its function [1]. Although aging is the primary cause, it is increasingly recognized as a multifactorial syndrome that may also develop in the context of chronic illnesses where systemic inflammation, metabolic dysregulation, and treatment-related effects accelerate muscle wasting [2]. While age-related muscle decline is a natural physiological process, in certain patients sarcopenia may progress at a markedly accelerated rate, leading to profound functional impairment, heightened morbidity, and increased mortality risk. Consequently, recognizing and addressing sarcopenia within patient management is of critical importance.

Sarcopenia has varying prevalence and risk factors across different populations. Globally, sarcopenia affects 10% to 16% of the elderly, with higher prevalence in patients with specific conditions, such as diabetes and cancer [3]. For example, in China, the prevalence among older adults (≥60 years) is approximately 20.7% [4]. In Egypt, a study focusing on older adults with fragility fractures found a sarcopenia prevalence of 69.7%, highlighting a strong correlation between sarcopenia risk and fracture risk [5]. In contrast, rural regions in India have demonstrated a notably higher prevalence of sarcopenia compared to urban areas, with contributing factors including older age, inadequate dietary protein intake, and lower socioeconomic status contributing to increased risk [6]. Understanding these variations is crucial for developing targeted interventions and public health strategies aimed at mitigating the impact of sarcopenia, particularly in vulnerable populations.

This review systematically examines the known causes, current definitions, and diagnostic approaches of sarcopenia, as well as recent advancements in its management (Figure 1). In addition, it discusses the emerging role of systems biology in identifying potential biomarkers and therapeutic targets for sarcopenia. Furthermore, we highlight how these approaches may contribute to the development of novel interventions aimed at enhancing muscle health and improving the quality of life in older adults.

The primary aim of this paper is to provide a comprehensive overview of the current state of knowledge on sarcopenia, to explore the emerging role of systems biology and its methodologies in advancing sarcopenia research, and to underscore the clinical relevance of systems biology by addressing the complex interplay between sarcopenia and other chronic diseases. The methodological framework for this review was based on a systematic literature search conducted across major academic databases, including PubMed (https://pubmed.ncbi.nlm.nih.gov/, accessed on 23 January 2025), Web of Science (https://www.webofscience.com/wos/, accessed on 23 January 2025), Scopus (https://www.scopus.com/, accessed on 23 January 2025), and Google Scholar (https://scholar.google.com/, accessed on 23 January 2025), up to the year 2025. Only articles published in English were included.

### 1.1. Pathophysiology of Sarcopenia

To develop effective solutions, it is crucial to understand the underlying mechanisms. In the case of sarcopenia, this involves examining the complex interplay between hormonal shifts, inflammation, dysfunctions of cellular and subcellular components, lack of nutrition, and many other influences, which eventually lead to a gradual decline in muscle mass and strength.

Previous studies have revealed that sarcopenia is associated with a reduction in both the size and number of type II muscle fibres, accompanied by an increase in intramuscular fat infiltration [18]. Type II muscle fibres, particularly type IIx and IIb, are more susceptible to atrophy with ageing, as seen in the extensor digitorum longus muscle, where these fibres show a significant reduction in cross-sectional area in older mice compared to younger ones [19]. Similarly, elderly female hip fracture patients exhibit extensive atrophy of type II muscle fibres compared to both healthy elderly and young individuals, highlighting the vulnerability of these fibres to age-related degeneration [20]. Conversely, the accumulation of intramuscular fat exacerbates sarcopenia by compromising muscle quality rather than merely reducing muscle mass through impairments in muscle contractility and overall functional performance [21]. The interplay of sarcopenia and sarcopenic obesity, which is characterized by increased fat mass and muscle wasting, underscores the complex metabolic and physiological changes in ageing muscles, with type II fibre atrophy being a critical factor in the increased risk of falls and fractures in the elderly [22].

Mitochondria play a key role in energy production and cellular homeostasis. In sarcopenia, their dysfunction is associated with impaired mitophagy, altered mitochondrial dynamics, and reduced biogenesis [23,24]. Studies demonstrated that its dysfunction represents one of the key pathophysiological pathways in the context of sarcopenia [25,26,27]. Moreover, the accumulation of reactive oxygen species (ROS) and the resultant oxidative stress further exacerbate mitochondrial damage, contributing to muscle atrophy [23]. Also, there is a notable dysregulation of satellite cell homeostasis in sarcopenia, which impairs their ability to effectively repair and regenerate muscle tissue [28,29]. This impairment is partly attributed to mitochondrial dysregulation [30]. This connection between mitochondrial dysfunction and the disease’s progression emphasizes the need for targeted treatments to improve mitochondrial function and reduce the effects of sarcopenia in older adults.

Inflammation is connected to several diseases related to aging and is a major factor in muscle loss in older individuals [31,32]. The immune microenvironment of skeletal muscle undergoes significant alterations in sarcopenia, with macrophages playing a pivotal role in mediating the inflammatory processes observed in aged muscle tissues [33]. These immune cells contribute to the chronic inflammation that exacerbates muscle deterioration. Also, the interaction between adipose tissue and skeletal muscle involves adipose tissue acting as an endocrine organ that releases pro-inflammatory molecules and contributes to the progression of sarcopenia [34]. Systemic inflammatory markers such as C-reactive protein (CRP) and interleukin-1β (IL-1β) are elevated in individuals with sarcopenia, indicating a strong association between inflammation and muscle loss [35,36]. Moreover, dietary factors that promote inflammation can exacerbate sarcopenia symptoms, highlighting the role of lifestyle and diet in managing this condition [31].

Ageing has a profound effect on the endocrine system, resulting in reduced levels of anabolic hormones such as testosterone, growth hormone (GH), and insulin-like growth factor 1 (IGF-1). These hormones play a vital role in muscle protein synthesis and maintenance, and their decline ultimately contributes to the development of sarcopenia [37,38]. Testosterone, for instance, plays a vital role in maintaining muscle mass and function, and its deficiency, along with poor nutrition and lack of exercise, is a modifiable contributor to sarcopenia [39]. Likewise, the decline in oestrogen levels in postmenopausal women can result in metabolic dysfunction and an increased risk of developing sarcopenia [18]. Similarly, GH and IGF-1 are associated with muscle mass, and their decline with age is linked to reduced muscle protein synthesis and increased muscle degradation [37,40]. The concept of “anabolic resistance” further complicates this scenario, where the muscle’s response to protein intake is blunted in older adults, leading to insufficient muscle protein synthesis despite adequate nutrition [41]. This resistance is exacerbated by factors such as obesity, sedentarism, and insulin resistance, which are prevalent in the ageing population [41].

Denervation and motor neuron loss, including the reduction in motor units (MUs) and neuromuscular junction (NMJ) instability, are other key factors in the pathophysiology of sarcopenia, as emphasized in previous studies [42,43,44]. This loss of motor neurons and subsequent denervation lead to muscle atrophy and weakness [45,46]. Research indicates that these neuromuscular impairments often precede clinically diagnosed sarcopenia, suggesting that monitoring MU properties and NMJ stability could be crucial in early detection and intervention [45,47]. On the other hand, several studies confirm that vascular aging, endothelial dysfunction, reduced capillary density, and impaired blood flow are significant contributors to sarcopenia [48,49]. These vascular impairments reduce the delivery of oxygen and nutrients to skeletal muscles, which reduces anabolism in myocytes, leading to muscle atrophy and impaired function.

The gut microbiome functions as a critical regulator in the pathogenesis of sarcopenia. Alterations in microbial diversity and composition have been closely associated with both the onset and progression of this condition through several interconnected mechanisms [50]. For instance, gut dysbiosis reduces the abundance of bile salt hydrolase–producing bacteria, resulting in the accumulation of primary bile acids [51]. This accumulation suppresses Farnesoid X receptor signalling, thereby impairing muscle protein synthesis and accelerating muscle atrophy [51]. Moreover, metabolites derived from the gut microbiota, such as short-chain fatty acids and branched-chain amino acids, play a supportive role in muscle metabolism by exerting anti-inflammatory effects and stimulating protein synthesis via the AMP-activated protein kinase signalling pathway [52]. Conversely, dysbiosis not only diminishes the production of these beneficial metabolites but also increases the generation of harmful metabolites, including indole and p-cresol, which further contribute to muscle degeneration [52].

Understanding the mechanisms behind the pathophysiology of sarcopenia is crucial for creating treatment strategies to prevent muscle loss, maintain functional independence in older adults, and ultimately decrease the prevalence of sarcopenia and its substantial burden on the healthcare system. We illustrated the pathophysiological changes associated with sarcopenia in Figure 2.

### 1.2. Molecular and Clinical Biomarkers of Sarcopenia

As discussed in the earlier section, sarcopenia has a complex underlying mechanism, making it essential to find reliable biomarkers for early diagnosis and treatment. Various studies have explored potential biomarkers, ranging from blood-derived markers to metabolic and genetic indicators. Identifying these biomarkers can lead to more personalized treatment approaches, enabling healthcare providers to diagnose sarcopenia more effectively and monitor its progression.

Since inflammation plays a key role in the pathophysiology of sarcopenia, several inflammatory markers have been associated with sarcopenia in previous research. Inflammation is a critical component in the pathophysiology of sarcopenia, with several inflammatory markers being associated with the condition. Research has identified elevated levels of pro-inflammatory cytokines such as interleukin-6 (IL-6), interleukin-8 (IL-8), tumor necrosis factor-alpha (TNF-α), and IL-1β in individuals with sarcopenia [53,54,55]. In addition, high levels of CRP are associated with losing muscle strength and are seen as a good indicator of inflammatory status for investigating sarcopenia [53,54]. In addition, complete blood cell counts derived inflammatory indicators, such as the neutrophil-to-lymphocyte ratio (NLR), neutrophil-monocyte to lymphocyte ratio (NMLR), and systemic inflammatory response index (SIRI), have been associated with a higher occurrence of sarcopenia and an increased risk of death, emphasizing their importance in predicting outcomes [56].

Beyond inflammatory markers, dp-ucMGP has shown promise as a biomarker, with lower levels associated with sarcopenia and reduced physical performance [57]. Coenzyme Q10 and muscle proteins like irisin and creatine kinase have also been suggested as potential biomarkers, with lower levels observed in sarcopenic individuals [58]. Likewise, increased levels of myostatin and growth differentiation factor-15 (GDF-15), negative regulators of muscle growth, were linked to muscle wasting, hence they might be good candidates as a biomarker [59]. Other studies employing metabolomic analysis demonstrated that phosphatidylinositol 32:1 and Cathepsin D exhibit high diagnostic accuracy for sarcopenia [60,61]. Seo et al. highlighted sphingolipid metabolites, particularly ceramides and sphingomyelins, as potential circulating biomarkers, especially in men [62]. Proteomic profiling of extracellular vesicles has identified platelet factor 4 (PF4) and complement C1r subcomponent (C1R) as novel biomarkers, with significant diagnostic power [63].

Furthermore, noncoding RNAs and specific genes, including Acss1, Mtfp1, and Oxct1, have been implicated in sarcopenia, providing insights into the molecular mechanisms underlying muscle loss [64,65]. Noncoding RNAs modulate signalling pathways that govern skeletal muscle physiology and dysfunction. For instance, miR-434-3p is downregulated in ageing muscle, which enhances the activity of eIF5A1 and subsequently induces cell death through the mitochondrial apoptotic pathway [66]. In parallel, Acss1 contributes to energy metabolism, Mtfp1 regulates mitochondrial dynamics and apoptosis, and Oxct1 is essential for ketone body utilization [64]. Dysregulated expression of these genes, together with noncoding RNA-mediated regulation, underscores their central role in skeletal muscle atrophy and highlights their potential as biomarkers and therapeutic targets in sarcopenia.

Although previous findings are promising, no single biomarker has yet achieved optimal diagnostic accuracy, highlighting the necessity for further research to validate these results and create robust diagnostic tools [67]. Therefore, a combination of serum markers, diagnostic imaging, and functional tests of muscle function is suggested as an ideal biomarker panel for sarcopenia [68,69].

### 1.3. Diagnosis

Sarcopenia is diagnosed using multiple frameworks developed by international groups (Table 1). The diversity in approaches reflects the absence of a universal diagnostic standard. Despite differences among frameworks, they generally include assessments of muscle strength, mass, and physical performance. The primary diagnostic methods commonly used in clinical practice include those developed by the European Working Group on Sarcopenia in Older People (EWGSOP2), the Asian Working Group for Sarcopenia (AWGS), the International Working Group on Sarcopenia (IWGS), and the Foundation for the National Institutes of Health (FNIH) Sarcopenia Project.

EWGSOP2 is a revised consensus on the definition and diagnosis of sarcopenia and is widely utilized across Europe [70]. It emphasizes low muscle strength, measured through tools like handgrip strength tests, as the primary criterion [70]. This is further supported by evaluations of muscle mass, using imaging techniques such as CT or MRI, and assessments of physical performance to determine severity. On the other hand, the AWGS modifies these criteria to account for the Asian population’s unique characteristics, incorporating culturally relevant thresholds for muscle mass and emphasizing gait speed as a critical performance metric [71]. Also, AWGS recommends using the SARC-F questionnaire, calf circumference measurements, and performance tests like the 5-time chair stand test to identify at-risk individuals [71]. This regional adjustment emphasizes the need to customize diagnostic tools for different populations to achieve improved predictive results [72].

IWGS defines sarcopenia by focusing on low muscle mass and poor physical performance, without considering muscle strength as a criterion [73]. This approach is particularly emphasized for bedridden patients or those with slow gait speed, necessitating careful body composition assessments [73]. Similarly, the FNIH Sarcopenia Project aimed to address the same issue by analysing existing data sources to define diagnostic criteria. Initial findings were shared in May 2012, with final recommendations published in 2014 [74]. The FNIH criteria prioritize gait speed assessment, followed by handgrip strength and lean mass evaluation via dual-energy X-ray absorptiometry. In contrast to EWGSOP2, AWGS, and IWGS, the FNIH recommended a BMI-adjusted muscle mass index rather than one adjusted for height [73].

Despite growing interest in sarcopenia, a universally accepted consensus on its diagnostic criteria has yet to be established. Although various international working groups have proposed differing diagnostic guidelines, this heterogeneity may lead to confusion and inconsistency in clinical practice. A universal, standardized approach is needed to balance specificity and global applicability. From this perspective, systems biology and multi-omics data integration can help to develop better diagnostic tools and tailor them to demographic and personal differences for better outcomes.

**Table 1 ijms-26-08527-t001:** A summary table is provided to present the commonly applied diagnostic tools for sarcopenia. The following abbreviations are used: SARC-F represents Strength, Assistance with walking, Rising from a chair, Climbing stairs, and Falls. SARC-CalF refers to the SARC-F questionnaire in combination with calf circumference measurement, while EBM stands for Elderly Body Mass Index. CT indicates Computed Tomography, and MRI denotes Magnetic Resonance Imaging. DXA corresponds to Dual-energy X-ray Absorptiometry, whereas BIA refers to Bioelectrical Impedance Analysis. EWGSOP represents the European Working Group on Sarcopenia in Older People, with EWGSOP2 being its updated framework. Similarly, AWGS refers to the Asian Working Group for Sarcopenia, and AWGS2 is the updated version. IWGS denotes the International Working Group on Sarcopenia. FNIH stands for the Foundation for the National Institutes of Health Sarcopenia Project, and SDOC refers to the Sarcopenia Definitions and Outcomes Consortium.

Diagnostic Approach	Description	Common Methods	References
Screening Questionnaires	Initial screening to assess risk and functional status	SARC-F, SARC-CalF, SARC-F+EBM	[7]
Muscle Mass Measurement	Quantifying muscle mass to confirm sarcopenia diagnosis	CT, MRI, DXA, BIA	[8]
Muscle Strength Assessment	Measuring muscle strength as an indicator of muscle function	Handgrip Strength Dynamometer, Chair Stand Test	[75]
Physical Performance Testing	Functional assessment of mobility and endurance	Gait speed test, Short Physical Performance Battery, Timed Up and Go	[7]
Composite Diagnostic Criteria	Integrates muscle mass, strength, and physical performance	EWGSOP, EWGSOP2, AWGS, AWGS 2, IWGS, FNIH, SDOC	[73]
Anthropometric Measures	Simple measurements as surrogates for muscle mass and nutritional status	Calf Circumference, Mid-upper Arm Circumference	[76]

### 1.4. Current Treatment Options

Sarcopenia currently lacks FDA-approved treatments, offering limited therapeutic options [77]. However, various conventional methods and innovative approaches are being studied [78]. These treatment options can be broadly divided into non-pharmacological interventions (Table 2) and pharmacological approaches (Table 3).

Resistance training and sufficient protein intake are fundamental non-pharmacological strategies for managing the disease, as they support the preservation of muscle mass and its function [77,79]. Resistance training is recommended as the main therapy for sarcopenia, as research has shown it can boost muscle strength and size, enhance functionality, and promote muscle development [80,81]. It is advised to work out twice a week and mix upper and lower body workouts each time to achieve adequate results [82]. Furthermore, sufficient protein intake, especially when paired with physical exercise, is important for controlling sarcopenia as well. It is recommended to consume more protein daily than the recommended dietary allowance (0.8 g of protein for every kilogram of body weight) to maintain muscle mass and achieve optimal results from resistance training [83].

Despite the limited availability of approved pharmacological treatments for sarcopenia, an array of therapeutic interventions is presently under investigation. Previous research has shown that testosterone, selective androgen receptor modulators (SARMs), and other hormonal treatments such as oestrogen and dehydroepiandrosterone (DHEA) can improve muscle mass and strength, but these gains have not consistently led to better physical performance [84]. Also, myostatin inhibitors have shown promising outcomes in previous studies, but additional research is required for validation [85]. Moreover, a new combination drug, RJx-01, which includes metformin and galantamine, has shown combined advantages in early studies, enhancing muscle quality, stabilizing neuromuscular junctions, and lowering overall inflammation [86]. Other compounds such as apelin and irisin are also being investigated for their potential benefits in sarcopenia treatment [85].

In addition to these pharmacological approaches, modulation of the gut microbiota has emerged as a promising therapeutic avenue. Interventions such as probiotics, prebiotics, faecal microbiota transplantation, and selective use of antibiotics to suppress detrimental microbial communities have been proposed to mitigate gut dysbiosis associated with sarcopenia [52]. Nonetheless, while these pharmacological and microbiome-based therapies are still under development, exercise and nutritional interventions remain the cornerstone of clinical management for sarcopenia [77]. Given the multifactorial and systemic nature of the disease, future research must prioritize integrative approaches, with systems biology playing a critical role in identifying optimal therapeutic strategies.

**Table 2 ijms-26-08527-t002:** A summary table of the non-pharmacological treatment approaches for sarcopenia, including exercise, dietary supplements, herbal medicines, and other interventions.

Treatment Type	Non-Pharmacological Approach	Description	Mechanism of Action	References
Exercise Interventions	Progressive Resistance Training	Structured weight-bearing exercises using progressive overload principles	Upregulates protein synthesis, increases type II muscle fibre size	[82]
	Low Resistance Training	Gentler form of resistance training suitable for frail elderly or those with joint limitations	Promotes muscle protein synthesis with reduced joint stress	[87]
	Blood Flow Restriction Training	Innovative technique combining low-intensity exercise with partial vascular occlusion using specialized cuffs	Metabolic stress-induced hypertrophy, enhanced protein synthesis	[87]
	Aerobic Exercise	Cardiovascular training including walking, cycling, swimming, or dancing	Improves cardiovascular function, enhances muscle oxidative capacity	[88]
	Combined Training	Multimodal approach integrating resistance training, aerobic exercise, balance, and flexibility components	Synergistic effects on multiple physiological systems	[88]
Nutritional Interventions	Protein Supplementation	High-quality protein intake to meet increased needs in older adults	Provides essential amino acids for muscle protein synthesis	[89]
	Leucine Supplementation	Branched-chain amino acid supplementation focusing on leucine, the primary trigger for muscle protein synthesis	Stimulates mTOR pathway, triggers muscle protein synthesis	[89]
	Essential Amino Acids	Complete amino acid supplementation providing all nine essential amino acids that cannot be synthesized by the body	Direct substrate for muscle protein synthesis	[89]
	Vitamin D Supplementation	Fat-soluble vitamin essential for muscle function and calcium homeostasis	Improves muscle fibre function, calcium homeostasis	[89]
	Omega-3 Fatty Acids	Anti-inflammatory fatty acids that support muscle health by reducing inflammation and potentially enhancing muscle protein synthesis response to exercise and protein intake	Anti-inflammatory effects, enhances muscle protein synthesis	[89]
Herbal Medicine and Natural Supplements	Curcumin	Active compound from turmeric with potent anti-inflammatory and antioxidant properties	Anti-inflammatory, antioxidant, promotes muscle regeneration	[90]
	Green Tea Extract	Polyphenol-rich extract containing epigallocatechin gallate, a powerful antioxidant that may protect against muscle atrophy and support muscle protein synthesis pathways	Antioxidant, enhances muscle protein synthesis	[91]
	Ginseng	Traditional adaptogenic herb used for centuries to combat fatigue and enhance physical performance	Adaptogenic, improves energy metabolism, anti-fatigue	[92]
	Astragalus membranaceus	Traditional Chinese medicine herb with immune-modulating properties	Immune modulation, muscle preservation	[93]
	Rhodiola rosea	Arctic root herb with adaptogenic properties that may help improve exercise capacity, reduce fatigue, and enhance recovery from physical stress in aging populations	Adaptogenic, anti-fatigue, improves physical performance	[94]
	Creatine Monohydrate	A supplement that increases muscle phosphocreatine stores, enabling rapid ATP regeneration during high-intensity activities	Increases phosphocreatine stores, enhances ATP regeneration	[95]
	β-Hydroxy β-Methylbutyrate	Metabolite of leucine with anti-catabolic properties	Reduces protein breakdown, anti-catabolic effects	[96]
Physical Therapy Approaches	Neuromuscular Electrical Stimulation	Therapeutic technique using electrical impulses to stimulate muscle contractions	Direct muscle fibre stimulation, protein synthesis activation	[97]
	Whole-Body Vibration	Platform-based therapy delivering mechanical vibrations to the entire body during standing or exercise positions	Enhances neuromuscular activation, bone-muscle interaction	[98]
	Balance Training	Exercises focusing on proprioception, stability, and postural control	Improves neuromuscular control, reduces fall risk	[99]
	Functional Training	Task-specific exercises that mimic activities of daily living	Task-specific muscle activation, functional improvement	[100]
Other Non-Pharmacological Interventions	Heat Therapy	Application of heat through saunas, hot baths, or heating pads to promote muscle recovery and adaptation	Increases heat shock proteins, improves muscle protein synthesis	[101]
	Soft Tissue Manipulation	Manual manipulation of soft tissues to improve circulation, reduce muscle tension, and enhance recovery	Improves circulation, reduces muscle tension	[102]
	Acupuncture	Traditional Chinese medicine technique involving insertion of fine needles at specific body points	Stimulates sensory nerves at acupoints, enhances muscle repair processes	[103]

**Table 3 ijms-26-08527-t003:** Proposed drugs for the treatment of sarcopenia, their descriptions, and development stages.

Drug Name	Description	Development Stage	References
**ACE-083**	a recombinant fusion protein for muscle growth	clinical research	[9]
**Bimagrumab**	a monoclonal antibody for activin receptor	clinical research	[104]
**Dehydroepiandrosterone**	an endogenous steroid hormone precursor	clinical research	[105]
**Enobosarm**	a selective androgen receptor modulator	clinical research	[10]
**Growth hormone**	a peptide hormone produced by the pituitary gland	clinical research	[106]
**GSK-2881078**	a selective androgen receptor modulator	clinical research	[10]
**Isomyosamine**	a synthetic derivative of tobacco plant alkaloids	clinical research	[107]
**Landogrozumab**	a monoclonal antibody to target myostatin	clinical research	[108]
**Ligandrol**	a selective androgen receptor modulator	clinical research	[10]
**LPCN 1148**	an oral androgen receptor agonist	clinical research	[109]
**MK-0773**	a selective androgen receptor modulator	clinical research	[11]
**Oestrogen**	the primary female sex hormone	clinical research	[15]
**Perindopril**	an angiotensin-converting enzyme inhibitor	clinical research	[16]
**RJx-01**	a combination of metformin and galantamine	clinical research	[14]
**Sarconeos**	a MAS receptor activator	clinical research	[110]
**Testosterone**	the primary male sex hormone	clinical research	[12]
**Trevogrumab**	a monoclonal antibody to target myostatin	clinical research	[111]
**Apelin**	a peptide hormone	preclinical research	[112]
**Irisin**	a hormone and myokine	preclinical research	[113]
**MG-132**	a cell-permeable proteasome inhibitor	preclinical research	[17]
**NT-1654**	a C-terminal fragment of mouse agrin	preclinical research	[114]
**Troglitazone**	an antidiabetic and anti-inflammatory drug	preclinical research	[17]
**Vorinostat**	a histone deacetylase inhibitor	preclinical research	[115]

### 1.5. Preventive Strategies

Preventive strategies are central to reducing both the occurrence and progression of sarcopenia, particularly in vulnerable populations such as older adults experiencing malnutrition or maintaining sedentary lifestyles. These strategies primarily emphasize physical activity, nutritional optimization, and integrated interventions aimed at preserving muscle mass, strength, and functional capacity [116]. Among these, progressive resistance exercise training has been identified as the most effective preventive measure, given its ability to significantly enhance muscle mass and strength [117]. Furthermore, multi-joint exercises are particularly recommended due to their functional relevance in older adults [75]. From a nutritional perspective, sufficient protein intake, especially from high-quality protein sources, play a critical role in mitigating muscle loss [89]. Additionally, supplementation with omega-3 fatty acids and vitamin D has shown potential benefits; however, evidence indicates that the most effective preventive approach lies in combining nutritional strategies with progressive resistance training, thereby maximizing synergistic effects on muscle health [89].

## 2. Systems Biology

Systems biology is an interdisciplinary field that integrates experimental and computational approaches to study biological systems holistically and comprehensively. It aims to understand the complex interactions within and across biological systems, from molecular to organismal levels, by employing large-scale measurement techniques and mathematical modelling [118,119]. This approach emphasizes a complete view of biology rather than just breaking it down into smaller parts. It examines living beings as whole units and focuses on traits that emerge when all the separate parts come together to form an organism. It has revolutionized the study of diseases by providing insights into the mechanisms of complex biological systems, such as metabolic networks and signalling pathways, which are crucial for understanding physiological responses and disease mechanisms [120,121]. Thus, systems biology approach is crucial for grasping the intricate relationships among metabolic pathways, genetic control systems, environmental influences, and lifestyle choices that lead to sarcopenia.

Systems biology became a unique field in the late 20th century, mainly due to improvements in high-throughput omics technologies and the need to examine large biological datasets. The creation of technologies allowed for extensive measurements of cellular functions, making it easier to study biological systems on a broad scale. Next, researchers combined computational techniques with biological studies, enabling a complete understanding of complex biological systems, such as gene expression and biological pathways.

### Methodologies

Systems biology is a multidisciplinary discipline that employs various methodologies to study complex biological systems (Figure 3). These methodologies encompass omics technologies, network analysis, computational modelling, and machine learning. Each of these approaches provides unique insights into the functioning of biological systems, enabling researchers to understand the intricate interactions and dynamics within these systems. By integrating these methodologies, systems biology aims to offer a comprehensive understanding of biological processes, which may facilitate advancements in personalized medicine, enhance diagnostic precision, and guide the development of targeted treatment strategies.

Omics technologies provide comprehensive data on different aspects of biological systems [122]. With the advanced technologies, we now have access to a vast array of omics data. The most used types include genomics, transcriptomics, proteomics, metabolomics, lipidomics and metagenomics, focusing on the in-depth analysis of genes, RNA, proteins, metabolites, lipids and the microbiome, respectively. These data types allow researchers to explore specific layers of the central dogma of molecular biology, aiding in the identification of key disease mechanisms. For instance, Ceyhan et al. investigated the transcriptomics data of patients with Collagen VI-related dystrophy, which led to the identification of potential target genes and candidate drugs for therapeutic intervention [123]. Furthermore, integrating data from multiple omics approaches has enabled researchers to achieve a more holistic understanding of biological systems, uncover intricate molecular interactions, and create more precise predictive models of cellular behaviour. For example, Meng et al. leveraged multi-omics data derived from patients with Alzheimer’s disease to identify key plasma proteins and gut microbiota species associated with disease severity prediction [124].

Network analysis is a powerful approach in systems biology for understanding complex multidimensional biological data. This methodology enables researchers to analyse interactions and relationships across multiple biological layers, from genes and proteins to metabolites and phenotypes [125]. Biological networks can represent diverse molecular interactions, such as metabolic pathways, gene co-expression patterns, protein–protein interactions, cell signalling pathways, and gene regulatory mechanisms. By analysing these networks, scientists can identify key components and interactions that drive biological processes, leading to a more comprehensive understanding of cellular functions and disease pathophysiology. Notably, Kaynar et al. investigated the gene co-expression network patterns of glioblastoma multiforme and identified significant modules and hub genes that may serve as potential therapeutic targets for treatment [126].

Computational modelling is a valuable technique that employs computers to simulate and analyses complex systems. In systems biology, this method has gained significant importance, especially using genome-scale metabolic models (GEMs). GEMs are network-based tools that integrate metabolic data, including genes, enzymes, reactions, and metabolites, with high-throughput biological data [127]. They simulate metabolic pathways, predict phenotypes, and help identify therapeutic targets, especially for human diseases and drug discovery [127]. The integration of machine learning with GEMs has further improved their predictive accuracy and efficiency, making them valuable tools in translational medicine and drug discovery [128]. In the context of the human gut microbiome, GEMs have been used to predict interactions between diet, microbes, and the host, providing insights into the role of microbiota in disease onset, such as cancer and metabolic disorders [129]. Although GEMs are useful, they encounter challenges such as oversimplification, limited access to experimental data and kinetic parameters, and inadequate representation of regulatory mechanisms [130,131]. However, advancements in genome sequencing and the development of automated reconstruction tools are expected to enhance the applicability and effectiveness of GEMs in various biological and medical research areas [128].

## 3. Integrating Systems Biology Approaches in Sarcopenia Studies

Systems biology has been used to discover better diagnostic and treatment options for various diseases. Its methodologies have become increasingly valuable in sarcopenia research as well, offering new insights into the molecular mechanisms of the condition and potential therapeutic targets. These approaches integrate various omics data types and analytical methods to provide a comprehensive understanding of sarcopenia’s complex pathophysiology.

Genomic studies have identified a range of genetic markers associated with sarcopenia, including 78 SNPs linked to sarcopenia and 55 SNPs related to sarcopenic obesity in the UK Biobank data [132]. Additionally, genetic biomarkers such as RPS10, NUDT3, and GPD1L, associated with lean body mass and appendicular skeletal muscle mass, have been identified in Asian populations [133]. Transcriptomics studies have further identified potential therapeutic targets. For example, Ceyhan et al. utilized RNA-seq data with network analysis to identify AGL and PDHX as possible targets for sarcopenia, suggesting MG-132 and troglitazone as pharmacological treatment candidates [17]. In the research done by Zhang and his team, the immune-related transcriptional regulatory network was explored, and the PAX5-SERPINA5-PI3K/Akt pathway was identified as a possible mechanism in sarcopenia [134].

Proteomic analysis is useful in understanding the complex interactions of proteins involved in muscle metabolism and degeneration. By employing mass spectrometry techniques, researchers can identify specific protein changes associated with sarcopenia, leading to insights into potential biomarkers for early diagnosis and intervention strategies. Ubaida-Mohien et al. investigated muscle proteomics of 58 young and old healthy subjects and found that in older individuals, there was a decrease in ribosomal energy metabolism-related proteins, such as the TCA cycle, mitochondrial respiration, and glycolysis [135]. Conversely, proteins associated with inflammation and alternative splicing were more prevalent [95]. Moreover, previous proteomic profiling studies demonstrated a shift from fast muscle fibre types to slower ones with ageing [136].

Metagenomics has emerged as a powerful tool in understanding the complex relationship between gut microbiota and sarcopenia. Recent studies utilizing shotgun metagenomic sequencing have provided insights into how alterations in gut microbiota composition and function may contribute to the disease. For instance, the Xiangya Sarcopenia Study identified changes in the gut microbiome composition and function, highlighting the potential of metagenomics to uncover specific microbial species and their functional capacities that are altered in sarcopenia [137]. Likewise, the altered microbiota in sarcopenia is associated with changes in metabolic pathways, such as a reduction in genes involved in short-chain fatty acids synthesis, which are crucial for muscle health and inflammation modulation [138]. While these studies highlight the potential of targeting the gut-muscle axis to mitigate the effects of sarcopenia, future research is needed to validate these findings in human studies and to explore the clinical relevance of these microbial changes in sarcopenia management [139].

### The Relationship Between Sarcopenia and Other Diseases

Sarcopenia is closely associated with various diseases due to common underlying processes. Systems biology methods have been key in discovering these links, showing that sarcopenia is not just a result of ageing but also occurs alongside several chronic illnesses. These studies emphasize the complicated interaction of genetic, molecular, and environmental elements that play a role in the emergence and advancement of sarcopenia with other diseases. For instance, Xu et al. used a systems biology approach to identify 114 shared DEGs between heart failure and sarcopenia, linked to growth factors, insulin secretion, and the cGMP-PKG pathway, including seven hub genes as potential biomarkers or therapeutic targets [140]. This integrative perspective highlights the potential for developing targeted interventions that address both sarcopenia and its comorbid conditions, ultimately improving patient outcomes through a more holistic approach to treatment.

As another example, the link between sarcopenia and respiratory diseases has been investigated in previous research. Wang et al. showed that chronic obstructive pulmonary disease and sarcopenia have shared risk factors such as smoking, inflammation, and oxidative stress, and both conditions involve systemic inflammation and genetic similarities that lead to muscle loss and decreased function [141]. Another study indicated that the activation of alpha-beta T cells and the control of lymphocyte death processes are common pathways linking sarcopenia and COVID-19, emphasizing that the relationship between these two conditions could go both ways, possibly forming a destructive cycle [142]. Recent evidence also indicates that enhancing muscle mass may function as a preventive strategy against gastroesophageal reflux disease and metabolic dysfunction-associated fatty liver disease. These associations underscore the critical role of metabolic disorder management in lowering the risk of comorbid conditions frequently linked to sarcopenia [143,144].

On the other hand, several studies demonstrated a causal link between certain autoimmune conditions and accelerated muscle loss. For example, patients with rheumatoid arthritis exhibit a significantly higher prevalence of sarcopenia compared to healthy control populations [145,146]. Similarly, mendelian randomization studies revealed a causal negative relationship between Type 1 Diabetes Mellitus (T1DM) and appendicular lean mass, as the chronic inflammatory state associated with T1DM contributes to muscle wasting and reduced muscle function [145]. Likewise, a previous prospective study involving 158 patients with inflammatory bowel disease reported a sarcopenia prevalence of 34.2%, with a significant association between sarcopenia and hospital admissions within a year [147]. The findings from these studies indicate that autoimmune diseases negatively impact muscle physiology and accelerate sarcopenia. This relationship certainly results in negative clinical effects and makes treatment and prognosis more difficult.

## 4. Future Perspectives

The use of systems biology approaches in sarcopenia research has provided important insights into the disease’s underlying mechanisms and potential therapeutic targets. Nevertheless, there are still several limitations and challenges to overcome in this field. A key challenge lies in integrating diverse omics data types, which still creates significant computational and analytical difficulties [148]. The large volume and complexity of multi-omics data make it challenging to extract meaningful biological insights and pinpoint the critical pathways involved in sarcopenia’s development. Likewise, the lack of standardized data collection and analytic methods can lead to variability in findings, which reduces trust in the field [149].

Numerous studies on sarcopenia have been conducted so far. Yet, many of them involve relatively small sample sizes, limiting their statistical power and generalizability of findings. Also, most of the studies rely on cross-sectional datasets, which may not capture the dynamic nature of sarcopenia progression or its response to interventions over time [150,151]. On the other hand, differences in how sarcopenia is defined and diagnosed result in variations in the prevalence of the disease [152,153,154], which makes the problem even more complicated to solve. It is partly attributed to technical challenges in assessing muscle mass and quality, the lack of early-stage detection, and arbitrary cut-off points in diagnostic criteria for different ethnic groups [70,155]. Despite the potential of systems biology in identifying biomarkers and treatment targets, clinical validation remains a major challenge, slowing the translation of discoveries into practice.

Looking ahead, future perspectives in sarcopenia research are promising. Although there is no FDA-approved drug for the treatment of sarcopenia yet, there are promising drugs under investigation in phase trials for sarcopenia, such as trevogrumab and selective androgen receptor modulators [84,156]. In addition, due to improvements in technology, genome sequencing and multi-omics analysis are becoming less expensive over time [157,158], which will eventually allow researchers to collect more samples from sarcopenia patients at a lower cost. The integration of multi-omics technologies with computational modelling within a systems biology framework provides opportunities to identify distinct molecular signatures and pathway alterations linked to sarcopenia. Such individualized molecular profiles may facilitate patient stratification into biologically defined subgroups, guiding the development of targeted interventions such as precision-based nutritional supplementation [159], exercise protocols tailored to metabolic responsiveness, or microbiome-centred therapies.

Furthermore, the incorporation of longitudinal omics datasets with clinical and lifestyle variables into predictive network models may allow earlier identification of individuals at risk, thus enabling preventive measures before clinically apparent muscle decline [160]. Additionally, increasing investments in biomedical databases [161] and hardware infrastructures [162] are expected to further accelerate personalized and precision medicine applications for sarcopenia and related conditions [163]. Ultimately, systems biology may transition sarcopenia management from generalized treatment paradigms toward precision medicine approaches, thereby improving therapeutic efficacy and health outcomes in heterogeneous aging populations.

## 5. Conclusions

Sarcopenia presents a significant challenge in geriatric healthcare, impacting muscle mass, strength, and overall quality of life in older adults. Its pathophysiology involves an interplay of multiple factors such as mitochondrial dysfunction, inflammation, hormonal changes, and neuromuscular impairments. Unfortunately, there is no international consensus on its diagnosis and treatment. Nevertheless, systems biology has proven valuable in sarcopenia research, facilitating biomarker discovery, improving diagnostic tools, and identifying potential pharmacologic interventions. As research progresses, these advancements may ultimately lead to improved muscle health and increased quality of life for the ageing population, addressing the growing burden of sarcopenia on healthcare systems worldwide.

## Figures and Tables

**Figure 1 ijms-26-08527-f001:**
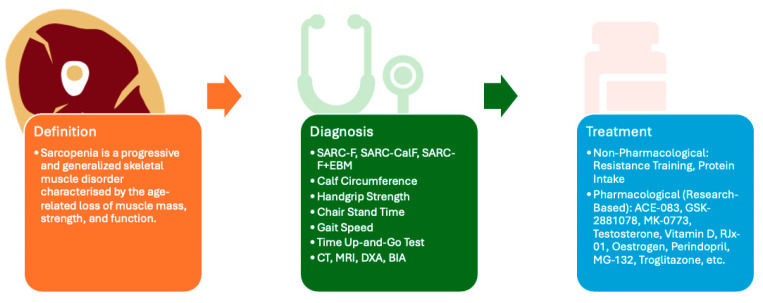
This figure provides an overview of the article, encompassing the definition, diagnostic approaches, and therapeutic strategies for sarcopenia. Although no universal consensus has been established regarding diagnostic criteria, commonly employed clinical assessment tools include SARC-F [7], SARC-CalF [7], SARC-F+EBM [7], calf circumference [7], handgrip strength [7], chair stand time [7], gait speed [7], time up-and-go test [7], computed tomography (CT) [8], magnetic resonance imaging (MRI) [8], dual-energy X-ray absorptiometry (DXA) [8], and bioelectrical impedance analysis (BIA) [8]. At present, no pharmacological treatment has received FDA approval; however, several clinical trials are investigating potential therapeutic agents, including ACE-083 [9], GSK-2881078 [10], MK-0773 [11], Testosterone [12], Vitamin D [13], RJx-01 [14], Oestrogen [15], Perindopril [16], MG-132 [17], and Troglitazone [17] among others.

**Figure 2 ijms-26-08527-f002:**
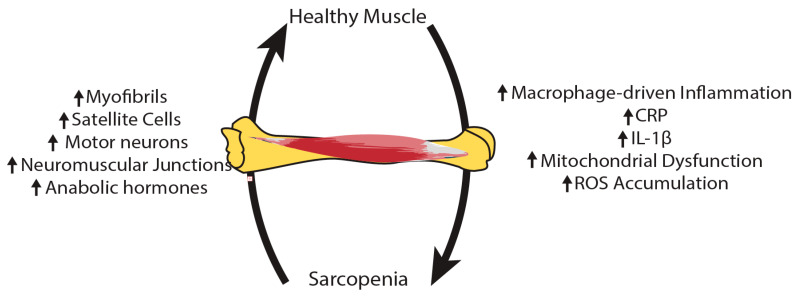
The pathophysiology of sarcopenia is complex and varied. As inflammation, mitochondrial dysfunction and reactive oxygen species (ROS) rise in sarcopenic muscles, there is a decline in the number of myofibrils, satellite cells, motor neurons, and neuromuscular junctions, along with anabolic hormones.

**Figure 3 ijms-26-08527-f003:**
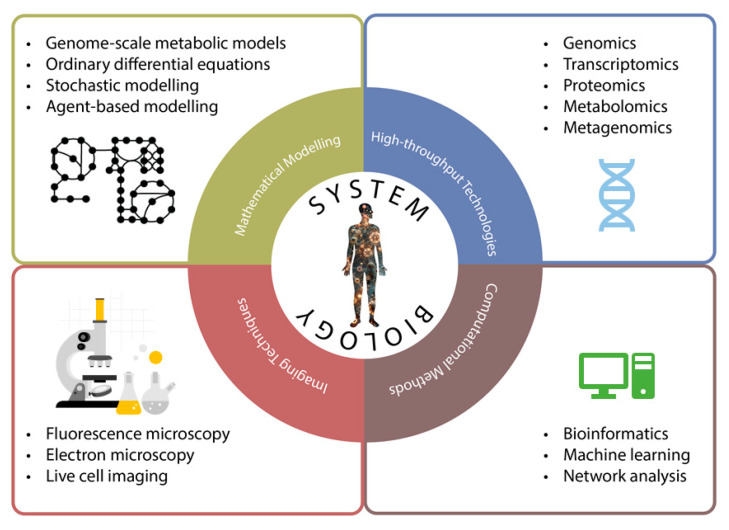
Systems biology adopts various methodologies from other disciplines to understand the complex nature of organisms. These methodologies can be mainly divided into four subtitles: high-throughput technologies, computational methods, mathematical modelling and imaging techniques.

## Data Availability

Not applicable.

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
