# Peer review of "Unravelling the Complexity of Sarcopenia Through a Systems Biology Approach"

_ijms, 2025, doi:10.3390/ijms26178527_

Round 1
Reviewer 1 Report
Comments and Suggestions for Authors
Unraveling the Complexity of Sarcopenia through Systems Biology Approach
Nice work, please address the below comments:
The author refers to sarcopenia as primarily due to aging which is one of the main causes but not the only with common reasons as cancer or with other chronic conditions. Widen defection with specification of the review scope will help shape the readers mind.
Figure1: there is some screening questionnaires could be added too. The authors includes treatment as if it is approved and effective for sarcopenia treatment which is not the case - you can add them as research-based approaches and the reference for each, reference needed for listed diagnosis criteria too.
Line 73: mitochondria is the most critical pathway - I think authors need to rephrase this as one of the critical or provide evidence for its top involvement- all the references provided demonstrate its involvement and not comparing it to other contributing factors.
Fig 2: is not showing the factors that the authors mentioned, it miss the mitochondria and immunological (macrophages role), there is non-mitochondrial roles or ROS not mentioned in text or the figure.
The vascular insufficiency or other pathology are not discussed too.
Line 154: noncoding RNAs and specific genes such as Acss1, Mtfp1, and Oxct1 have been implicated in sarcopenia, offering insights into mitochondrial dysfunction and muscle wasting- could you elaborate more how this ncRNAs link mitochondrial dysfunction and sarcopenia.
I liked table 1 for proposed treatment with their references, could the authors generate similar table for diagnosis combining all approaches presented in the literature and referring to the respective references.
I am suggesting also having table for non-pharmacological treatment approaches- beside exercise and protein intake- you can add the herbal medicine supplement and others- having all as a table with references would make it more enriched for readers.
Line 279- author added the word ref. at the end of the sentence instead of the real reference- kindly add it.
Figure 4: I feel conservative on having a figure for unpublished data specifically if no experimental approaches are reported- author can wave for it in the text as preliminary data though.
Line 343: Similar to how a tree's shadow reflects its shape on the ground, diseases reflect the composition of the microbiome within the human body. Metagenomics has emerged as a powerful tool in understanding the complex relationship between gut microbiota and sarcopenia, a condition characterized by the loss of muscle mass and function in older adults- are we still defining sarcopenia, I think this part is repeated and redundant.
Thanks
Author Response
Comments 1: The author refers to sarcopenia as primarily due to aging which is one of the main causes but not the only with common reasons as cancer or with other chronic conditions. Widen defection with specification of the review scope will help shape the reader’s mind.
Response 1: Thank you for your constructive feedback. To address this, we made this added these sentences in abstract and introduction sections below:
“While age-related muscle decline represents the most common form, sarcopenia may also develop as a secondary condition associated with chronic diseases, including cancer, diabetes, chronic obstructive pulmonary disease, and autoimmune disorders.”
“Although aging is the primary cause, it is increasingly recognized as a multifactorial syndrome that may also develop in the context of chronic illnesses where systemic inflammation, metabolic dysregulation, and treatment-related effects accelerate muscle wasting. While age-related muscle decline is a natural physiological process, in certain patients sarcopenia may progress at a markedly accelerated rate, leading to profound functional impairment, heightened morbidity, and increased mortality risk. Consequently, recognizing and addressing sarcopenia within patient management is of critical importance.”
Comments 2: Figure1: there is some screening questionnaires could be added too. The authors include treatment as if it is approved and effective for sarcopenia treatment which is not the case - you can add them as research-based approaches and the reference for each, reference needed for listed diagnosis criteria too.
Response 2: We are very grateful for your insightful comment. We have changed Figure 1 accordingly, and added respective references to figure legends, which shown below:
“This figure provides an overview of the article, encompassing the definition, diagnostic approaches, and therapeutic strategies for sarcopenia. Although no universal consensus has been established regarding diagnostic criteria, commonly employed clinical assessment tools include SARC-F135, SARC-CalF135, SARC-F+EBM135, calf circumference135, handgrip strength135, chair stand time135, gait speed135, time up-and-go test135, computed tomography (CT)136, magnetic resonance imaging (MRI)136, dual-energy X-ray absorptiometry (DXA)136, and bioelectrical impedance analysis (BIA)136. At present, no pharmacological treatment has received FDA approval; however, several clinical trials are investigating potential therapeutic agents, including ACE-083114, GSK-2881078 118, MK-0773123, Testosterone128, Vitamin D130, RJx-01126, Oestrogen124, Perindopril125, MG-13285, and Troglitazone85 among others.”
Comments 3: Line 73: mitochondria is the most critical pathway - I think authors need to rephrase this as one of the critical or provide evidence for its top involvement- all the references provided demonstrate its involvement and not comparing it to other contributing factors.
Response 3: Thank you for pointing out that fact. We rephrased our sentence as “Studies demonstrated that its dysfunction represents one of the key pathophysiological pathways in the context of sarcopenia.”
Comments 4: Fig 2: is not showing the factors that the authors mentioned, it miss the mitochondria and immunological (macrophages role), there is non-mitochondrial roles or ROS not mentioned in text or the figure.
Response 4: We have edited the Figure 2 according to your comment and updated its legend.
“The pathophysiology of sarcopenia is complex and varied. As inflammation, mitochondrial dysfunction and reactive oxygen species (ROS) rise in sarcopenic muscles, there is a decline in the number of myofibrils, satellite cells, motor neurons, and neuromuscular junctions, along with anabolic hormones.”
Comments 5: The vascular insufficiency or other pathology are not discussed too.
Response 5: Thank you for your comment. To address it, we added new sentences to article. As shown below:
“On the other hand, several studies confirm that vascular aging, endothelial dysfunction, reduced capillary density, and impaired blood flow are significant contributors to sarcopenia37,38. These vascular impairments reduce the delivery of oxygen and nutrients to skeletal muscles, which reduces anabolism in myocytes, leading to muscle atrophy and impaired function.”
Comments 6: Line 154: noncoding RNAs and specific genes such as Acss1, Mtfp1, and Oxct1 have been implicated in sarcopenia, offering insights into mitochondrial dysfunction and muscle wasting- could you elaborate more how this ncRNAs link mitochondrial dysfunction and sarcopenia.
Response 5: Thank you for underlining this missing point in our article. We have changed this line and create a new paragraph which elaborates more about how these ncRNAs and genes link mitochondrial dysfunction and sarcopenia.
“Furthermore, noncoding RNAs and specific genes, including Acss1, Mtfp1, and Oxct1, have been implicated in sarcopenia, providing insights into the molecular mechanisms underlying muscle loss50,51. Noncoding RNAs modulate signaling pathways that govern skeletal muscle physiology and dysfunction. For instance, miR-434-3p is downregulated in ageing muscle, which enhances the activity of eIF5A1 and subsequently induces cell death through the mitochondrial apoptotic pathway52. In parallel, Acss1 contributes to energy metabolism, Mtfp1 regulates mitochondrial dynamics and apoptosis, and Oxct1 is essential for ketone body utilization50. Dysregulated expression of these genes, together with noncoding RNA-mediated regulation, underscores their central role in skeletal muscle atrophy and highlights their potential as biomarkers and therapeutic targets in sarcopenia.”
Comments 7: I liked table 1 for proposed treatment with their references, could the authors generate similar table for diagnosis combining all approaches presented in the literature and referring to the respective references.
Response 7: We are grateful for your positive feedback. An extra table was added to the article related to for the diagnosis of sarcopenia combining the main approaches presented in the literature along with their references.
Comments 8: I am suggesting also having table for non-pharmacological treatment approaches- beside exercise and protein intake- you can add the herbal medicine supplement and others- having all as a table with references would make it more enriched for readers.
Response 8: Thank you for your feedback. We have added a new table after Table 1 to address your comment.
Comments 9: Line 279- author added the word ref. at the end of the sentence instead of the real reference- kindly add it.
Response 9: Thank you for your feedback. We have added the reference for it.
Comments 10: Figure 4: I feel conservative on having a figure for unpublished data specifically if no experimental approaches are reported- author can wave for it in the text as preliminary data though.
Response 10: We have removed the Figure 4 from the article.
Comments 11: Line 343: Similar to how a tree's shadow reflects its shape on the ground, diseases reflect the composition of the microbiome within the human body. Metagenomics has emerged as a powerful tool in understanding the complex relationship between gut microbiota and sarcopenia, a condition characterized by the loss of muscle mass and function in older adults- are we still defining sarcopenia, I think this part is repeated and redundant.
Response 11: We are grateful for your comment. The redundant part was removed.
Reviewer 2 Report
Comments and Suggestions for Authors
One of the key strengths of the manuscript is its comprehensive and detailed literature review, which systematically addresses the diagnostic approaches to sarcopenia, current therapeutic options, and the potential role of systems biology in advancing understanding of the disease. Particularly commendable is the authors’ effort to go beyond traditional perspectives by incorporating modern, integrative research directions, such as omics technologies, network analysis, and machine learning-based methodologies.
The logical flow of the introduction could be improved by first providing a structured overview of the background literature to establish the relevance and importance of the topic. Only after this should graphical elements (e.g. Figure 1) be introduced.
In addition, the objectives of the study should be clearly defined, and the methodological approach, including selection criteria, search strategies, inclusion/exclusion parameters... should be presented in a more structured and transparent manner, as this section is currently underdeveloped.
Additional content considerations to enrich the manuscript:
- It would be valuable to explore how the gut microbiome contributes to the pathogenesis of sarcopenia and how these mechanisms could be modulated through nutritional or pharmacological interventions.
- Preventive strategies: a discussion of effective preventive measures in aging populations - especially among high-risk groups such as those with malnutrition or sedentary lifestyles - would add important clinical relevance.
- Personalized approaches: further elaboration on how systems biology could support the development of individualized prevention or treatment strategies would enhance the manuscript’s innovative contribution.
The manuscript offers a valuable contribution to the field of sarcopenia research, particularly through its focus on systems-level approaches to unravel the disease’s complex pathophysiology. Incorporating the suggested revisions and additional content would enhance the scientific rigor and practical relevance of the article. Inclusion of more critical reflection, as well as summary tables (e.g. drug development stages, diagnostic criteria comparisons), would further support reader comprehension and improve the overall impact of the work.
Author Response
Comment 1: One of the key strengths of the manuscript is its comprehensive and detailed literature review, which systematically addresses the diagnostic approaches to sarcopenia, current therapeutic options, and the potential role of systems biology in advancing understanding of the disease. Particularly commendable is the authors’ effort to go beyond traditional perspectives by incorporating modern, integrative research directions, such as omics technologies, network analysis, and machine learning-based methodologies. The logical flow of the introduction could be improved by first providing a structured overview of the background literature to establish the relevance and importance of the topic. Only after this should graphical elements (e.g. Figure 1) be introduced.
Response 1: Thank you for your comment. We have added a new paragraph to illustrate a structured overview of the article, which is shown below:
“This review systematically examines the known causes, current definitions, and diagnostic approaches of sarcopenia, as well as recent advancements in its management (Figure 1). In addition, it discusses the emerging role of systems biology in identifying potential biomarkers and therapeutic targets for sarcopenia. Furthermore, we highlight how these approaches may contribute to the development of novel interventions aimed at enhancing muscle health and improving the quality of life in older adults.”
Comment 2: In addition, the objectives of the study should be clearly defined, and the methodological approach, including selection criteria, search strategies, inclusion/exclusion parameters... should be presented in a more structured and transparent manner, as this section is currently underdeveloped.
Response 2: We are grateful for your comment. To address your suggestion, we have added an additional paragraph to the introduction section, which is provided below:
“The primary aim of this paper is to provide a comprehensive overview of the current state of knowledge on sarcopenia, to explore the emerging role of systems biology and its methodologies in advancing sarcopenia research, and to underscore the clinical relevance of systems biology by addressing the complex interplay between sarcopenia and other chronic diseases. The methodological framework for this review was based on a systematic literature search conducted across major academic databases, including PubMed, Web of Science, Scopus, and Google Scholar, up to the year 2025. Only articles published in English were included.”
Comment 3: It would be valuable to explore how the gut microbiome contributes to the pathogenesis of sarcopenia and how these mechanisms could be modulated through nutritional or pharmacological interventions.
Response 3: Thank you for your feedback. We have included new paragraph to pathophysiology and treatment subsections of to address your comment.
“The gut microbiome functions as a critical regulator in the pathogenesis of sarcopenia. Alterations in microbial diversity and composition have been closely associated with both the onset and progression of this condition through several interconnected mechanisms39. For instance, gut dysbiosis reduces the abundance of bile salt hydrolase–producing bacteria, resulting in the accumulation of primary bile acids40. This accumulation suppresses Farnesoid X receptor signalling, thereby impairing muscle protein synthesis and accelerating muscle atrophy40. Moreover, metabolites derived from the gut microbiota, such as short-chain fatty acids and branched-chain amino acids, play a supportive role in muscle metabolism by exerting anti-inflammatory effects and stimulating protein synthesis via the AMP-activated protein kinase signalling pathway41. Conversely, dysbiosis not only diminishes the production of these beneficial metabolites but also increases the generation of harmful metabolites, including indole and p-cresol, which further contribute to muscle degeneration41.”
“In addition to these pharmacological approaches, modulation of the gut microbiota has emerged as a promising therapeutic avenue. Interventions such as probiotics, prebiotics, faecal microbiota transplantation, and selective use of antibiotics to suppress detrimental microbial communities have been proposed to mitigate gut dysbiosis associated with sarcopenia41.”
Comment 4: Preventive strategies: a discussion of effective preventive measures in aging populations - especially among high-risk groups such as those with malnutrition or sedentary lifestyles - would add important clinical relevance.
Response 4: We have added a new section to the article for preventive strategies.
“1.5 Preventive Strategies
Preventive strategies are central to reducing both the occurrence and progression of sarcopenia, particularly in vulnerable populations such as older adults experiencing malnutrition or maintaining sedentary lifestyles. These strategies primarily emphasize physical activity, nutritional optimization, and integrated interventions aimed at preserving muscle mass, strength, and functional capacity75. Among these, progressive resistance exercise training has been identified as the most effective preventive measure, given its ability to significantly enhance muscle mass and strength76. Furthermore, multi-joint exercises are particularly recommended due to their functional relevance in older adults76. From a nutritional perspective, sufficient protein intake, especially from high-quality protein sources, play a critical role in mitigating muscle loss 77. Additionally, supplementation with omega-3 fatty acids and vitamin D has shown potential benefits; however, evidence indicates that the most effective preventive approach lies in combining nutritional strategies with progressive resistance training, thereby maximizing synergistic effects on muscle health77.”
Comment 5: Personalized approaches: further elaboration on how systems biology could support the development of individualized prevention or treatment strategies would enhance the manuscript’s innovative contribution.
Response 5: Thank you for your comment. We have updated further perspectives section of the review.
“Looking ahead, future perspectives in sarcopenia research are promising. Although there is no FDA-approved drug for the treatment of sarcopenia yet, there are promising drugs under investigation in phase trials for sarcopenia, such as trevogrumab and selective androgen receptor modulators72,117. Besides, due to improvements in technology, genome sequencing and multi-omics analysis are becoming less expensive over time118,119, which will eventually allow researchers to collect more samples from sarcopenia patients at a lower cost. The integration of multi-omics technologies with computational modelling within a systems biology framework provides opportunities to identify distinct molecular signatures and pathway alterations linked to sarcopenia. Such individualized molecular profiles may facilitate patient stratification into biologically defined subgroups, guiding the development of targeted interventions such as precision-based nutritional supplementation120, exercise protocols tailored to metabolic responsiveness, or microbiome-centred therapies.
Furthermore, the incorporation of longitudinal omics datasets with clinical and lifestyle variables into predictive network models may allow earlier identification of individuals at risk, thus enabling preventive measures before clinically apparent muscle decline121. Additionally, increasing investments in biomedical databases122 and hardware infrastructures123 are expected to further accelerate personalized and precision medicine applications for sarcopenia and related conditions124. Ultimately, systems biology may transition sarcopenia management from generalized treatment paradigms toward precision medicine approaches, thereby improving therapeutic efficacy and health outcomes in heterogeneous aging populations.”
Comment 6: The manuscript offers a valuable contribution to the field of sarcopenia research, particularly through its focus on systems-level approaches to unravel the disease’s complex pathophysiology. Incorporating the suggested revisions and additional content would enhance the scientific rigor and practical relevance of the article. Inclusion of more critical reflection, as well as summary tables (e.g. drug development stages, diagnostic criteria comparisons), would further support reader comprehension and improve the overall impact of the work.
Response 6: We sincerely appreciate your feedback. In response to the suggestions provided by Reviewer 1 and Reviewer 2, we have incorporated new sections, expanded the content, and included additional tables in our manuscript.